# Error Enhancement for Upper Limb Rehabilitation in the Chronic Phase after Stroke: A 5-Day Pre-Post Intervention Study

**DOI:** 10.3390/s24020471

**Published:** 2024-01-12

**Authors:** Marjan Coremans, Eli Carmeli, Ineke De Bauw, Bea Essers, Robin Lemmens, Geert Verheyden

**Affiliations:** 1Department of Rehabilitation Sciences, KU Leuven, 3001 Leuven, Belgium; inekedebauw@gmail.com (I.D.B.); bea.essers@kuleuven.be (B.E.); 2Department of Physical Therapy, University of Haifa, Haifa 3498838, Israel; ecarmeli@univ.haifa.ac.il; 3Department of Neurosciences, Experimental Neurology, KU Leuven, 3000 Leuven, Belgium; robin.lemmens@kuleuven.be; 4Department of Neurology, University Hospitals Leuven, 3000 Leuven, Belgium

**Keywords:** stroke, upper limb rehabilitation, error enhancement

## Abstract

A large proportion of chronic stroke survivors still struggle with upper limb (UL) problems in daily activities, typically reaching tasks. During three-dimensional reaching movements, the deXtreme robot offers error enhancement forces. Error enhancement aims to improve the quality of movement. We investigated clinical and patient-reported outcomes and assessed the quality of movement before and after a 5 h error enhancement training with the deXtreme robot. This pilot study had a pre-post intervention design, recruiting 22 patients (mean age: 57 years, mean days post-stroke: 1571, male/female: 12/10) in the chronic phase post-stroke with UL motor impairments. Patients received 1 h robot treatment for five days and were assessed at baseline and after training, collecting (1) clinical, (2) patient-reported, and (3) kinematic (KINARM, BKIN Technologies Ltd., Kingston, ON, Canada) outcome measures. Our analysis revealed significant improvements (median improvement (Q1–Q3)) in (1) UL Fugl–Meyer assessment (1.0 (0.8–3.0), *p* < 0.001) and action research arm test (2.0 (0.8–2.0), *p* < 0.001); (2) motor activity log, amount of use (0.1 (0.0–0.3), *p* < 0.001) and quality of use (0.1 (0.1–0.5), *p* < 0.001) subscale; (3) KINARM-evaluated position sense (−0.45 (−0.81–0.09), *p* = 0.030) after training. These findings provide insight into clinical self-reported and kinematic improvements in UL functioning after five hours of error enhancement UL training.

## 1. Introduction

Good upper limb (UL) motor function is needed for daily life activities [1]; therefore, regaining UL function is often a priority for the stroke survivor [2]. However, almost half of the people after a stroke have contralesional UL deficits that restrict UL activities [3] and remain present even after six months post-stroke [3,4]. In this chronic phase after stroke, spontaneous recovery is no longer observed, motor recovery plateaus and motor function remains lower than before the stroke [5,6,7,8]. However, there is still potential for enhancing UL motor function through exercise-dependent plasticity using high-dose therapy [9,10]. Rehabilitation in the chronic phase thus remains important, and in order to achieve these high doses, robotic UL rehabilitation seems promising.

Recently, the use of robotic UL rehabilitation has become more widespread as it has several advantages [11,12,13,14]. Firstly, the number of movement repetitions can be increased safely and can be automatically captured. Several studies have shown a dose-response relationship, indicating that more repetitions result in greater motor recovery benefits [11,15,16]. Secondly, current literature shows the effectiveness of robot-based treatment in addition to conventional therapy, improving motor function [17,18] and enhancing motor learning [19,20]. Thirdly, robots enable the assessment of kinematic movement correlates, providing a means to evaluate the quality of movement [21]. Assessing the movement quality is important to understand improvements in UL capacity post-therapy, as recommended by the Stroke Recovery and Rehabilitation Roundtable [21]. 

One way to improve motor function and movement quality is through robot-based error enhancement. When a person performs a movement and deviates from the intended path, the robot will enlarge this error by applying external forces. As a result, the person will try to counter this error-driven disturbance, prompting them to strengthen their control [22]. As movement error plays an important role in learning, magnifying this error will likely stimulate this learning process [23,24], resulting in a refinement of movement coordination [25]. In addition, people after a stroke often have an impaired nervous system that is less sensitive to error and hence does not react to small errors. Augmentation of errors might make them noticeable and increase the likelihood that the patient will learn from them [24]. Besides, training with error enhancement is a form of implicit learning [26]. Implicit learning might be more feasible for patients after a stroke as it aims to minimize the involvement of cognitive resources [27]. This is what differentiates robot-based error enhancement from other robot-based rehabilitation. 

Robot-based error enhancement has recently been investigated in reaching studies [22,25,28]. Reaching is important for activities in daily life but is a common problem in people after stroke [29]. Reaching movements are less smooth and appear with more variability and an abnormal speed profile compared to healthy individuals [30]. This is where robot-based error enhancement can help. In healthy participants, error enhancement was shown to increase the accuracy of reaching movements [22,28]. In people after a stroke, a systematic review provides the first evidence of the effectiveness of this new method on UL motor impairment [31]. One study in a group of 26 chronic stroke participants reported an improvement in clinical outcomes [32], and another showed a positive effect on patient-reported outcomes [33]. In a group of 18 chronic stroke participants, improvements in a range of kinematics were identified [34]. Studies with the DeXtreme prototype (BioXtreme Ltd., Petah Tikva, Israel) revealed an improvement in movement error in healthy individuals [22] and movement smoothness in a stroke population [35]. While most studies included either observation-based clinical or kinematic outcomes to evaluate the effect of training on motor performance, the combination of both outcome measures was rare and only one study included a patient-reported outcome [31]. However, the use of patient-reported outcomes is important as they can reveal deficits in many patients with stroke that are not detected using observation-based assessments [36]. Other studies included in the systematic review had small sample sizes, limited training time, or lacked a control group, resulting in inconclusive results. Lastly, most studies focused on two-dimensional movements in the horizontal plane, whereas functional reaching movements are nearly always conducted three-dimensionally (3D). 

Therefore, we designed a pilot study in the chronic phase post-stroke using the DeXtreme robot (BioXtreme Ltd., Israel) that allows error enhancement during 3D reaching movements. We examined the effects of this novel robotic training approach with standardized (1) clinical measures, (2) patient-reported outcomes, and (3) kinematic measures of UL function. We hypothesised that after five hours of error enhancement training whereby participants would perform on average more than 1000 reaching movements, patients would (1) improve on clinical measures [22,28,31,35,37], (2) report better arm use in daily life [33], and (3) improve movement quality as measured with kinematics [33,34].

## 2. Materials and Methods

### 2.1. Participants

Adults with chronic stroke participated in this pilot study. They were recruited from our database and the discharge records of the University Hospitals Leuven Rehabilitation Center Pellenberg. In addition, we encouraged first-line general practitioners and physiotherapists to inform potential participants.

The inclusion criteria were (1) first-ever stroke, (2) minimum six months after stroke, (3) maximum 85 years old, and (4) a UL motor impairment, yet no severe stiffness: having less than 66 points (maximum) on the Fugl–Meyer assessment [38] for the UL (FMA-UE) but being able to open and close the hand five times, and bend and extend the elbow two times. Exclusion criteria were as follows: (1) sensory aphasia (item 9 of the National Institutes of Health Stroke Scale [39]: ≤2/3); (2) apraxia (apraxia screen of TULIA [40]: <9/12); (3) neglect (star cancellation test [41]: <44/54); (4) having a cognitive deficit (as defined by mini-mental state examination [42]: ≤24/30); or (5) the presence of shoulder pain in rest or during active shoulder movements. Given the exploratory nature of this study, a sample size calculation was not possible, yet we are convinced that a homogeneous group of 22 patients in the stable chronic phase after stroke will be able to inform us about the sample needed for subsequent research. 

### 2.2. Procedure

#### 2.2.1. Apparatus

During the study, the DeXtreme robot (BioXtreme Ltd., Israel) was used. The DeXtreme, an FDA- and CE-registered device, is a robotic arm that requires the patient to actively perform reaching movements in a three-dimensional workspace. During the reaching movements, the robot exerts error enhancement forces on the UL to magnify the errors. This end-effector robot (Figure 1) focuses on the facilitation of accuracy, range of movement, stability, and smoothness of UL movements. We used the updated version of the apparatus as described in a previous paper by Israely et al. [22], with updated features published by Carmeli et al. [43].

#### 2.2.2. Treatment Session Protocol

For this study, a pre- and post-intervention design was used. The total protocol duration was seven consecutive weekdays, starting with a pre-intervention assessment on day one, followed by five one-hour training sessions on five consecutive weekdays, and concluding with a post-intervention assessment on day seven. The study was conducted between January 2022 and November 2022 and obtained ethical approval from the Ethics Committee Research of KU/University Hospitals Leuven, Belgium (registration number: B3222021000614, internal ref. nr: S65699).

One training session lasted one hour and consisted of two blocks of twenty-minute robot training, alternated with an active break (stretching and auto-mobilization). The patient was seated in a chair placed in a standardised position and restrained with seatbelts to prevent trunk compensation movements. Before each training session, the robot was calibrated. Afterwards, the system was adjusted to the patient, requiring the patient to bring the arm to 90° anteflexion and fully extend the elbow while holding the gimbal. Anti-gravitation support could be offered according to the needs of the patient.

On average, one can do 12–20 reaching movements per game, and about 12 games can be played per therapy session. This results in about 192 movements per therapy session, and a total (average) of 960 reaching movements over 5 days.

#### 2.2.3. Games—Force Field Algorithm

During the robot training, two games were played: (1) the Market Stand, which focused on the range of motion and the accuracy of the movement (Figure 2a), and the Alchemist game, which emphasized stability and smoothness of movement (Figure 2b). Algorithms provided progression in terms of accuracy, range of movement, stability, and smoothness, depending on the performance of the patient. Each training session began with a game without error enhancement forces to establish the participant’s baseline. When needed, feedback was given by the therapist. Feedback was offered verbally, e.g., “Try to fully extend the elbow”, or tactile by guiding the patient once in the right direction. On-screen information in both games provided real-time feedback about the successfulness of the movement performed. 

The force field algorithms used differed between the two games. The algorithm applied during the Market Stand was perpendicular to the straight trajectory line and at a distance from it. Additionally, the forces decreased as the hand moved away from the starting position and increased as the error magnitude increased. During the Alchemist game, the force field algorithm was aligned with the direction of the movement of the hand, increasing the error of the acceleration component. A detailed description of the games and the algorithms used can be found in previous publications [22,43].

### 2.3. Outcome Measures

Demographic and health information was collected pre-intervention, including age, gender, working status, time after and type of stroke, lateralization of symptoms, and pre-stroke hand dominance. During pre- and post-intervention, we collected a battery of reliable and valid clinical, patient-reported, and kinematic measurements.

#### 2.3.1. Clinical Measurements

ICF body function level: UL motor impairment was assessed with the upper extremity subscale of the Fugl–Meyer Assessment (FMA-UE) [38,44,45,46]. A lower score indicates a more severe impairment. The first international stroke recovery and rehabilitation roundtable on measuring sensorimotor outcome agreed that the FMA-UE is the recommended UL motor function outcome for stroke recovery trials [47]. Two visual analogue scales (VAS) were used [48] to evaluate the pain and stiffness that the patient feels in the most affected UL, through visualization on a 10 cm line on paper [49]. Zero, on the left end of the line, represented no pain or stiffness. Both scores were converted to a score of 100. The VAS was not only assessed at pre- and post-intervention but also before and after every therapy session. Lastly, the motor assessment scale (MAS) for tone (MAS-tone) assessed muscle tonus [50,51]. Scores higher than 4 indicate persistent hypertonicity.

ICF activity level: An action research arm test (ARAT) evaluated the functional performance of the UL [46,52]. Higher values represent better performance. The first international stroke recovery and rehabilitation roundtable on measuring sensorimotor outcomes agreed that the ARAT is the recommended UL activity outcome for stroke recovery trials [47]. Additionally, the 7-point motor assessment scale [50,51,53] for the UL (MAS-UE) assessed everyday motor skills. On both scales, a higher score represents better performance.

#### 2.3.2. Patient-Reported Measurements

The hand subscale of the stroke impact scale (SIS) [54] evaluated patients’ perceptions of difficulties in using the affected hand to perform five activities of daily living. The total score is converted to a 100-point scale and a higher score indicates a better perceived performance. In addition, the amount of use (MAL-AOU) and quality of the movements (MAL-QOL) of the UL during daily living tasks were measured by the upper-extremity Motor Activity Log-14 items (MAL) [55]. The MAL-14 is a structured interview of 14 questions. A higher score indicates a higher amount and quality of use of the affected UL.

#### 2.3.3. Kinematic Measurements of Sensorimotor Function

The KINARM robot (BKIN Technologies Ltd., Kingston, ON, Canada) was used to test sensorimotor impairment. This bimanual end-point robot allows 2D movements in the horizontal plane. The virtual reality screen permits the control of visual feedback. Tests with the robot are performed in a seated position, with seatbelts to restrain trunk movements and a black cloth to prevent the vision of the arms. If needed, hand fixation was provided.

To test motor function, the 4-target visually guided reaching (VGR) test was performed with the affected arm. The patients were instructed to move the cursor to a red dot as accurately and fast as possible. Ten outcome parameters were calculated, including reaction time, speed, and accuracy of reaching. All the parameters were combined into a single task score with higher values meaning worse motor function [56,57].

A 4-target arm position-matching (APM) test was performed to assess the proprioception (position sense) of the affected arm. The robot brought the most affected arm into a position and the patient must actively move the less affected arm into the same position but mirrored, without any form of visual feedback. Twelve outcome parameters were calculated, covering variability and magnitude of position errors, and combined into a single task score with higher values meaning worse proprioception [57,58]. Both tests show good validity and reliability in participants with stroke [56,58,59]. Dexterit-E Explorer (version 3.9.3) was used to obtain the parameters of the VRG and the APM test.

Lastly, to test sensory processing, the discrimination task (DT) was performed. The patient was instructed to move the most affected arm and track down a 3-, 4-, or 5-angle figure, delineated by virtual walls, which were not visible to the patient. In the next step, the patient had to draw the same figure with the less affected arm without mirroring. Visual feedback was provided on the hand position. Finally, the patient had to identify the explored figure out of six options. A more detailed description of this task is described elsewhere and was found to be valid for people in the chronic phase after stroke [60]. To analyse the parameters of the DT, Dexterit-E Explorer (version 3.9.3) and MATLAB (version R2022b) were used [60]. Five parameters were calculated and combined in one-factor score, as proposed by Saenen et al. [60].

### 2.4. Statistical Analysis

Normality was checked for all variables with the Shapiro–Wilk test. Mean and standard deviations (SD) were calculated for normally distributed variables, medians with first quartile (Q1) and third quartile (Q3) for non-normally distributed variables or ordinal scales. Normally distributed variables were compared pre-post through parametric analysis (paired *t*-test) and not-normally distributed variables or ordinal scales through non-parametric analysis (Wilcoxon signed ranks test). Data were analysed with IBM SPSS Statistics for Windows, version 28.0 (IBM Corp, Armonk, NY, USA) with the level of two-tailed statistical significance set at *p* < 0.05. For all non-parametric analyses, effect sizes were calculated with r and interpreted as small (r = 0.1), medium (r = 0.3), and large (r = 0.5). For all parametric analyses, effect sizes were calculated with G_Hedges_, and interpreted as small (G = 0.2) medium (G = 0.5) and large (G = 0.8). As this was a pilot study, an exploratory data analysis was conducted without correction for multiple tests.

## 3. Results

### 3.1. Participant Characteristics

We recruited 22 patients and Table 1 reports the patient characteristics for the demographic and general stroke-related variables. Our sample included 12 women and 10 men with a mean age of 57 years. The mean days since stroke for the total group were 1571 days (range: 184–11, 751 days), showing that we recruited mostly people who experienced their stroke several years ago. For 12 patients, their right UL was most affected. Most patients (N = 20) were right-handed pre-stroke.

Two patients dropped out because of adverse effects, one after the second, and one after the third therapy session. The first patient reported increased pain and tension in the neck-shoulder line and headaches, and the other reported increased stiffness in the hand. Both patients were followed up, and the complaints were resolved but the two patients decided not to continue with the study. Thus, we present analyses based on results from 20 patients.

### 3.2. Clinical Outcomes

Clinical results pre- and post-intervention are presented in Table 2. The median (IQR) FMA-UE value pre-treatment was 54 (50–58) out of 66 points and the median (IQR) ARAT was 50 (39–53) out of 57 points, demonstrating that our sample included people with moderate to mild UL motor impairment. Results from pre- to post-intervention analyses for the clinical variables are also presented in Table 2. A significant pre- to post-intervention improvement was found for UL function measured with FMA-UE (median (IQR) improvement of 1.0 (0.8–3.0) points, *p* < 0.001), and UL activity assessed with ARAT (median (IQR) improvement of 2.0 (0.8–2.0) points, *p* < 0.001), with large effect sizes (r ≥ 0.5). There were no significant pre- to post-intervention differences for VAS and MAS. A detailed overview of the score per participant can be found in Appendix A.

### 3.3. Patient-Reported Outcomes

Patient-reported results pre- and post-intervention and analyses are presented in Table 3. A significant pre- to post-intervention improvement was found for the self-perceived amount of UL use evaluated by MAL-AOU (median (IQR) improvement of 0.1 (0.0–0.3) points, *p* < 0.001), and perceived quality of movement investigated by MAL-QOM (median (IQR) improvement of 0.1 (0.1–0.5) points, *p* < 0.001), with large effect sizes (r = 0.5). There were no significant pre- to post-intervention differences for the SIS hand. A detailed overview of the score per participant can be found in Appendix B.

### 3.4. Kinematic Outcomes

Results of the kinematic variables collected pre- and post-intervention are presented in Table 4. For the arm position matching task, a significant change was observed in absolute error in the X direction (*p* = 0.048, r = 0.3) with an error reduction in the frontal plane when matching arm positions with the less affected upper limb when the robot offers these positions to the more affected UL. Also, for the arm position matching composite task score (*p* = 0.03, r = 0.3), a significant improvement was noted in the overall performance on this task. We found medium effect sizes for both outcomes. For the visually guided reaching, only the posture speed change, the median hand speed when the hand should be at rest, reached significance (*p* < 0.001, G = 0.9) with a greater speed registered at the end of the visually guided reaching protocol. For the other variables and the composite task score of the discrimination test, no significant differences were found. A detailed overview of the score per participant can be found in Appendix C.

### 3.5. Number of Reaching Movements during 5 h of Error Enhancement Training

Table 5 presents the number of reaching movements participants performed during the 5-day intervention protocol. The mean (SD) amount was 1043 (127) reaching movements with a minimum of 723 and a maximum of 1236 movements.

## 4. Discussion

Our study investigated the hypothesis that five one-hour sessions on five consecutive days of reaching training incorporating error enhancement would provide clinical, patient-reported, and kinematic improvements in chronic stroke survivors with residual UL impairments and activity limitations. Our results support this postulation, as we observed improvements in UL motor function and capacity, perceived upper limb performance, and position sense through kinematic evaluation. 

Clinically, we demonstrated significant improvements in UL motor function, UL activity and self-perceived performance, as measured by the Fugl-Meyer assessment for the upper extremity, action research arm test and motor activity log amount of use and quality of movement, respectively. Although the improvements are rather small, they are noteworthy given the relatively limited duration of our intervention. While five hours of therapy is rather limited, it is important to note that time in training may not accurately reflect training intensity [61]. The number of repetitions performed during this training is a more accurate indicator of training intensity [62], and we found that our participants on average performed 1043 reaching repetitions during the five-hour training period. Moreover, as the active error enhancement training time was only 40 min per hour, we argue that our sample performed many reaches within the available time, making our intervention of interest for further consideration in clinical research and practice. These findings are especially relevant when considering the provision of training in the chronic phase of stroke recovery and add to the existing body of knowledge in this domain. Despite the high number of repetitions of reaching movements, no patient experienced an increase in pain or stiffness in the UL during or after the therapy sessions. This is consistent with previous research which included intensive upper limb rehabilitation [9,63,64,65].

Our study has several notable strengths. First and foremost, our intervention focused on the provision of a high number of reaching movements, providing a concentrated and targeted approach to UL rehabilitation. Additionally, our protocol was comprehensive, including clinical, self-reported and kinematic evaluation. This allowed us to obtain a robust understanding of the efficacy of our intervention and provided a strong foundation for future research in this area. Moreover, the clinical outcomes were well-established and widely accepted as important measures of recovery and rehabilitation in stroke survivors. This consensus-based methodology [47,66] ensures that our findings are not only relevant for research purposes but also have practical implications for clinical practice. Another important finding was the absence of a negative impact of five one-hour sessions performed every day for one week, as revealed by no changes in the visual analogue scale for tone and pain in the UL. While two participants did experience increased tension and pain and ultimately chose to discontinue the study, it is important to note that one participant was already suffering from increased tension in the neck-shoulder line before baseline, and the other had a history of increased stiffness in the UL. Based on this experience, we would consider refining our inclusion criteria for future studies to ensure that individuals with pain or increased muscle tension are more carefully screened. While we did observe significant improvements in clinical outcomes at the ICF body function (FMA-UE) and activity level (ARAT), we did not observe changes in the motor assessment scale (MAS), also at the activity level. The pre-score on the MAS was already high (median: 14/18), which may explain the limited progress. Moreover, there is some debate about the hierarchy of items in this scale; the ranking of items seems inconsistent, as some patients can complete the most difficult task, but fail an easier item [67,68]. There was no improvement in the self-reported stroke impact hand subscale. However, we did see an improvement in the motor activity log, another self-report assessment. We believe that the MAL may be a more relevant outcome measure for our intervention than SIS-Hand, which focuses specifically on hand function and may be limited in its ability to capture improvements in overall limb function. 

Our study also included kinematic UL evaluation, which showed a significant improvement in arm position matching, indicating better position sense after the intervention. This improvement may be attributed to the error enhancement component of the training, which provided increased somatosensory input during reaching movements, which may have resulted in better somatosensory awareness [69,70], reflected in a better position-sense outcome. It would have been of interest to see whether this improvement could also have been present when position sense was tested clinically but our protocol did not include standard clinical somatosensory evaluation. We did not find any significant changes in visually guided reaching and discrimination tasks, which may be due to the task-specific nature of these evaluations. The discrimination task evaluates movement sense and sensory discrimination, while sensory discrimination was not an element included in our training protocol. Surprisingly, we observed a worsening in posture speed, which evaluates the stability of the UL before and after reaching, when the hand should be at rest. This may be linked to anticipation during the reaching training when performing the reaches. When re-evaluating reaching with a kinematic task after training, the training anticipation may have reflected in moving quicker, however, this could have increased posture speed. There are several potential explanations for the lack of improvement in kinematic reaching. Firstly, our evaluation protocol involved two-dimensional reaching while our task involved three-dimensional reaching, which may have impacted learning. Secondly, reaching kinematics is considered a parameter of the quality of movement and may reflect restitution [47,71], which is unlikely to occur in the chronic phase after stroke. Lastly, the provided intensity may not have been sufficient to induce kinematic changes. The average number of repetitions was 1043, which may still not be enough for detecting kinematic changes in motor control of reaching. It would be interesting to investigate in larger rehabilitation trials whether the improvements found are correlated between the different levels of the ICF.

Some limitations of our study must be acknowledged. One limitation is that the average age of our sample was younger than the usual age of people after a stroke in Belgium [72]. The literature, however, shows that age has limited influence on motor recovery after stroke on long-term outcome measures [73,74]. Another limitation is the large range in time after stroke of our participants. However, all patients were in the chronic stage after stroke and there is currently no research indicating that long-term after stroke, movement adaptability may alter. Our intervention protocol was also limited in duration, consisting of only one hour per day for five days and no follow-up measurement. However, our main focus was on the number of reaches participants would perform, and we reached our proposed target with an average of 1043 repetitions. Further studies should include a follow-up measurement to investigate the sustainability of the improvements. In addition, our protocol included a two-dimensional kinematic analysis, while our intervention trained three-dimensional reaching. A three-dimensional reaching task would be beneficial to include in future studies to better evaluate the quality of UL movements. Unfortunately, the availability of technology is a limiting factor in study development. Therefore, a three-dimensional drinking task to evaluate the quality of UL movement is recommended by the Second Stroke Recovery and Rehabilitation Roundtable [21]. Moreover, a reaching task consists not only of reaching but also includes a distal manipulation component. During our therapy, different manipulation components were not practised, therefore outcome measures that specifically measure hand function were not used. Further studies should incorporate this. Furthermore, this study did not include a blinded assessor, thus we cannot exclude assessor bias for the clinical outcomes. Further studies with a blind assessor would be beneficial to strengthen the validity of our results. Finally, we did not include a control group or correct for multiple testing due to the exploratory nature of the study.

In summary, our study suggests that an hourly intervention for five days which actively stimulates reaching movements through serious gaming with error enhancement might improve UL function, capacity, and self-reported UL performance in people in the chronic phase after stroke with mild residual impairments in UL function and activity. Further research should investigate these findings in a larger experimental group and include a control group in which reaching tasks are performed in null field conditions and with a blinded assessor. Thereafter, further work can expand on these findings by integrating the therapy concept in an overall UL treatment package for people in the chronic phase after stroke to improve the quality of movement post-stroke.

## Figures and Tables

**Figure 1 sensors-24-00471-f001:**
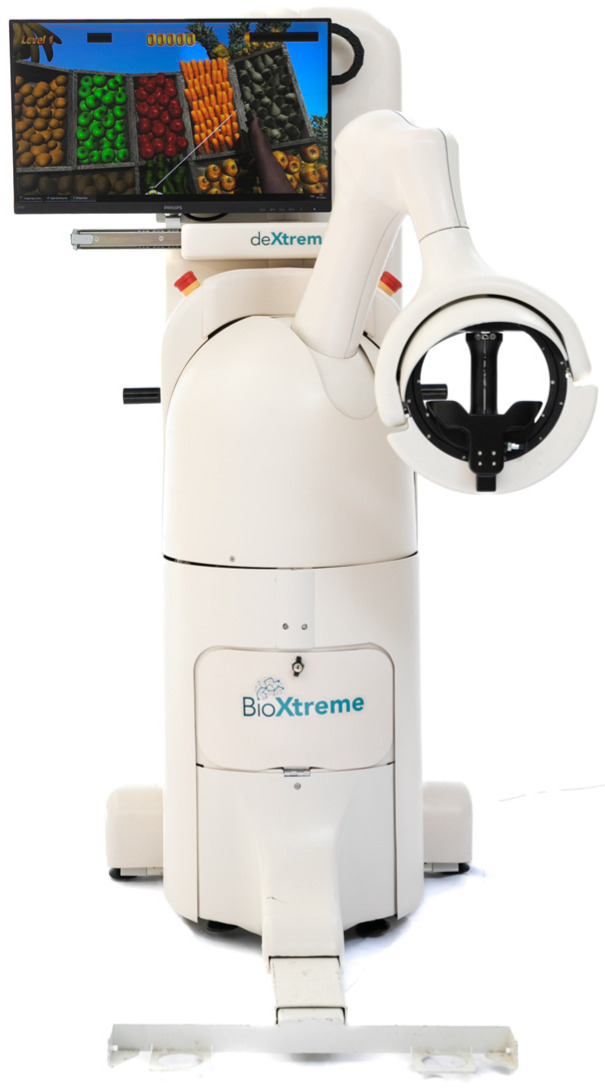
DeXtreme robot (BioXtreme Ltd., Israel).

**Figure 2 sensors-24-00471-f002:**
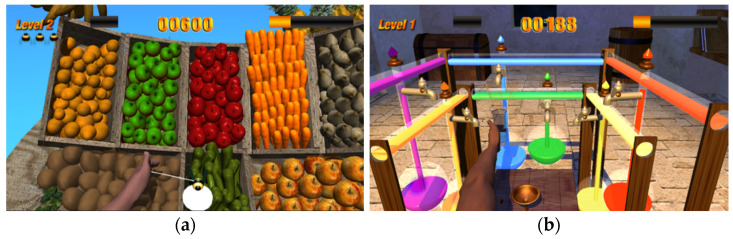
DeXtreme games using error enhancement forces during 3D reaching movements: (**a**) Market Stand game: the subject must follow the trajectory of a bee as quickly and accurately as possible. The bee moves from the starting point (white circle) to a random fruit box; (**b**) Alchemist game: the subject must fill a glass of water and move towards the coloured tap that lights up randomly, without spilling the water.

**Table 1 sensors-24-00471-t001:** Demographic and stroke-related characteristics are presented as mean and range (grey), or number in % (white).

Subject (N = 22)	Age	Days since Stroke Onset	Gender (M/F)	Work Status	Stroke Ethology	Most Affected Arm	Hand Dominance	Dominant Hand = Most Affected Side
1	57	297	M	Full time	Ischemia	Right	Right	Yes
2	64	2003	M	Retirement	Ischemia	Right	Right	Yes
3	65	1166	F	Retirement	Ischemia	Right	Right	Yes
4	69	621	M	Retirement	Ischemia	Left	Right	No
5	70	845	M	Retirement	Ischemia	Right	Right	Yes
6	33	1662	M	Fulltime	Ischemia	Right	Left	No
7	49	11,751	M	Fulltime	Bleeding	Left	Right	No
8	59	1201	F	Invalidity	Ischemia	Left	Right	No
9	57	196	F	Parttime	Ischemia	Left	Right	No
10	71	2869	M	Retirement	Ischemia	Right	Right	Yes
11	54	1113	F	Fulltime	Ischemia	Left	Right	No
12	71	2201	F	Retirement	Bleeding	Right	Right	Yes
13	57	680	F	Invalidity	Bleeding	Left	Right	No
14	56	979	F	Invalidity	Bleeding	Right	Right	Yes
15	65	184	M	Retirement	Bleeding	Right	Right	Yes
16	44	306	M	Invalidity	Ischemia	Right	Left	No
17	66	1580	F	Retirement	Ischemia	Left	Right	No
18	60	1219	M	Fulltime	Ischemia	Left	Right	No
19	28	907	F	Invalidity	Bleeding	Left	Right	No
20	45	571	F	Parttime	Ischemia	Left	Right	No
21	56	910	M	Invalidity	Ischemia & bleeding	Right	Right	Yes
22	51	1307	M	Invalidity	Bleeding	Right	Right	Yes
**Mean (SD)**	57 (12)	1571 (2372)	M: 55%	Working: 32%	Ischemia: 66%	Right: 55%	Right: 91%	Dominant side affected: 45%
**Range**	28–71	184–11,751	F: 45%	Not Working: 68%	Bleeding: 34%	Left: 45%	Left: 9%	Non-dominant side affected: 55%

SD: standard deviation; M: Male; F: Female.

**Table 2 sensors-24-00471-t002:** Clinical outcomes pre- and post-intervention presented as median (IQ1–IQ3).

Outcome Parameter	Median (IQ1–IQ3)		Median Difference (IQ1–IQ3)	*p*-Value	Effect Size (r)
	PRE	POST			
Fugle-Meyer Assessment—UE ^a^	54.0 (50.0–57.8)	55.0 (51.3–59.5)	1.0 (0.8–3.0)	<0.001 *	0.5
Visual Analogue Scale—tone ^a^	15.5 (5.8–30.0)	6.0 (1.3–30.0)	−5.0 (−13.0–0.2)	0.089	0.3
Visual Analogue Scale—pain ^a^	0.0 (0.0–4.5)	0 (0.0–1.8)	0.0 (−1.5–0.3)	0.178	0.2
Action Research Arm Test ^a^	49.5 (39.3–53.0)	50.5 (41.3–55.0)	2.0 (0.8–2.0)	<0.001 *	0.6
Motor Assessment Scale—tone ^a^	4.0 (4.0–4.8)	4.0 (4.0–4.8)	0.0 (0.0–0.0)	1.000	0.0
Motor Assessment Scale—UE ^a^	14.0 (12.0–15.5)	14.0 (11.3–16.3)	0.0 (0.0–0.0)	0.317	0.2

UE: Upper Extremity; IQ1–IQ3: interquartile range; ^a^: Wilcoxon signed-rank test; * *p* < 0.05.

**Table 3 sensors-24-00471-t003:** Patient-reported outcomes pre- and post-intervention presented as median (IQ1–IQ3).

Outcome Parameter	Median (IQ1–IQ3)		Median Difference (IQ1–IQ3)	*p*-Value	Effect Size (r)
	PRE	POST			
Motor activity log—AOU ^a^	1.9 (1.5–2.5)	2.1 (1.6–2.7)	0.1 (0.0–0.3)	<0.001 *	0.5
Motor activity log—QOM ^a^	1.9 (1.3–2.5)	2.1 (1.5–3.1)	0.1 (0.1–0.5)	<0.001 *	0.5
Stroke Impact Scale—Hand ^a^	57.5 (32.5–75.0)	65.0 (35.0–75.0)	0.0 (0.0–6.3)	0.837	0.0

AOU: the amount of use; QOM: quality of movement; IQ1–IQ3: interquartile range; ^a^: Wilcoxon signed-rank test; * *p* < 0.05.

**Table 4 sensors-24-00471-t004:** Kinematic outcome measures pre- and post-intervention presented as mean (SD) or median (IQ1–IQ3).

Kinematic Test	Outcome Parameter	Mean (SD) Median (IQ1–IQ3)		Mean Difference (SD) Median Difference (IQ1–IQ3)	*p*-Value	Effect Size (r/G)
		PRE	POST			
Arm Position Matching	Absolute Error X ^b^	0.048 (0.039–0.065)	0.0388 (0.032–0.053)	−0.005 (−0.021–(−0.001))	0.048 *	0.3
	Absolute Error Y ^b^	0.029 (0.025–0.037)	0.025 (0.019–0.042)	−0.002 (−0.009–0.003)	0.502	0.1
	Absolute Error XY ^b^	0.063 (0.051–0.076)	0.0505 (0.041–0.064)	−0.009 (−0.019–0.000)	0.052	0.3
	Variability X ^a^	0.039 (0.015)	0.033 (0.009)	−0.006 (0.015)	0.099	0.4
	Variability Y ^b^	0.016 (0.012–0.019)	0.015 (0.011–0.019)	−0.002 (−0.003–0.003)	0.526	0.1
	Variability XY ^a^	0.043 (0.017)	0.037 (0.010)	−0.006 (0.015)	0.105	0.4
	Contraction/expansion ratio X ^a^	0.937 (0.317)	0.906 (0.249)	−0.031 (0.217)	0.524	0.1
	Contraction/expansion ratio Y ^b^	0.980 (0.845–1.042)	1.014 (0.890–1.081)	0.024 (−0.042–0.075)	0.526	0.1
	Contraction/expansion ratio XY ^a^	0.949 (0.424)	0.925 (0.342)	−0.024 (0.273)	0.696	0.1
	Shift X ^a^	−0.001 (0.043)	−0.001 (0.044)	0.000 (0.034)	0.973	0.0
	Shift Y ^a^	−0.019 (0.022)	−0.021 (0.025)	−0.002 (0.019)	0.678	0.1
	Shift XY ^b^	0.042 (0.031–0.061	0.035 (0.022–0.053)	−0.004 (−0.016–0.007)	0.391	0.1
	Task Score ^b^	1.753 (1.145–2.066)	1.199 (0.460–1.853)	−0.448 (−0.806–0.092)	0.030 *	0.3
Visually Guided Reaching	Posture Speed ^a^	0.005 (0.005)	0.006 (0.005)	0.002 (0.002)	<0.001 *	0.9
	Reaction Time ^b^	0.335 (0.302–0.354)	0.325 (0.301–0.391)	0.009 (−0.014–0.028)	0.287	0.2
	Initial Direction Angle ^a^	0.066 (0.049)	0.071 (0.070)	0.005 (0.037)	0.582	0.1
	Initial Distance Ratio ^a^	0.769 (0.215)	0.764 (0.225)	−0.004 (0.126)	0.881	0.0
	Initial Speed ratio ^a^	0.973 (0.042)	0.958 (0.080)	−0.015 (0.074)	0.378	0.2
	Speed Maxima Count ^a^	2.844 (1.365)	2.757 (1.427)	−0.088 (1.011)	0.703	0.1
	Min Max Speed Difference ^b^	0.020 (0.010–0.027)	0.016 (0.013–0.027)	−0.002 (−0.005–0.001)	0.156	0.2
	Movement Time ^a^	1.320 (0.447)	1.237 (0.365)	−0.083 (0.350)	0.303	0.2
	Path Length Ratio ^a^	1.195 (0.219)	1.179 (0.174)	−0.016 (0.075)	0.359	0.2
	Max Speed ^b^	0.193 (0.169–0.259)	0.212 (0.186–0.239)	0.011 (−0.003–0.030)	0.279	0.2
	Task Score ^b^	2.998 (1.472–4.203)	3.499 (1.639–3.909)	0.061 (−0.571–0.662)	0.823	0.0
Discrimination Task	Factor Score ^a^	−0.232 (1.278)	−0.306 (1.191)	0.074 (1.001)	0.746	0.1

SD: standard deviation, IQ1–IQ3: interquartile range, ^a^: paired *t*-test, effect size (G_Hedges_); ^b^: Wilcoxon signed-rank test, effect size (r); * *p* < 0.05.

**Table 5 sensors-24-00471-t005:** Number of reaching movements during 5 h of error enhancement training.

Subject (N = 20)	Number of Repetitions
1	1061
2	935
3	1002
4	1028
5	723
6	852
7	1026
8	1236
9	1122
10	1024
11	1142
12	951
13	1125
14	1183
15	1116
16	939
17	1052
18	963
19	1204
20	1170
**Mean (SD)**	1043 (127)
**Range**	723–1236

SD: standard deviation.

## Data Availability

The data analysed during the current study are available from the corresponding author, Marjan Coremans, upon reasonable request.

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
