# Peer review of "Error Enhancement for Upper Limb Rehabilitation in the Chronic Phase after Stroke: A 5-Day Pre-Post Intervention Study"

_sensors, 2024, doi:10.3390/s24020471_

Round 1

Reviewer 1 Report

Comments and Suggestions for Authors

This manuscript entitled “Error-enhancement for upper limb rehabilitation in the chronic phase after stroke: a 5-day pre-post intervention study” was primarily aimed to explore the effects of the error-enhancement training on upper limb rehabilitation. The authors bring an interesting study, but there are still some problems that cannot up this article to a publishing level. Suggestions are listed in the specific comments below.

Specific comments:

1.     In the Abstract part, line 17, “recruiting 22 patients in the chronic phase post-stroke with UL motor impairments” please detailed anthropometry information for patients, such as gender information, height, weight.

2.     In the Introduction part, line 40-41, “Recently, the use of robotic UL rehabilitation has become more widespread as it has several advantages.” More references are required here to support this statement.

3.     In the Introduction part, line 55, “…control. [19].” Please delete the extra full stop.

4.     In the Results part, Participant Characteristics, in my opinion, it would be more appropriate to put the participant information in the Materials and Methods part.

5.     In the Discussion part, it is recommended to provide a brief description of the aim and main findings in the first paragraph of the manuscript.

6.     In the Discussion part, line 330, “…and provides a strong foundation…” please replace “provides” with “provided”.

7.     In the Discussion part, line 342-347, “While we did observe… the ranking of the items seems inconsistent [60,61].” This statement is too general and unclear. Can you be more specific?

8.     Some recently studies could be added in the discussion, such as:

The Effect of Application of Asymmetry Evaluation in Competitive Sports: A Systematic Review. Physical Activity and Health, 6(1), p.257–272.

Clinical Applications of Virtual Reality in Musculoskeletal Rehabilitation: A Scoping Review. Healthcare 2023, 11, 3178.

9.     Did authors calculate the sample size before the experiment? Is it enough to support the findings?

Comments on the Quality of English Language

Moderate editing of English language required

Reviewer 2 Report

Comments and Suggestions for Authors

In a study involving 22 chronic stroke survivors, a five-hour error-enhancement training for upper limbs resulted in:

  • Enhanced clinical assessments (FMA-UE, ARAT)
  • Improved patient-reported measures (MAL)
  • Better KINARM-assessed position sense.

This training showed significant improvements in upper limb functionality for these individuals post-stroke.

This is an interesting work. Following are my comments:

-          Abstract should be rewritten by illustrating the backgrounds and the main contributions

-          Line 79: please cite the related study.

-          Line 86: please add the link between previous works and their limitations against the proposed work illustrating the main contributions.

-          Lines 102-110: selection criteria of the participants are clear. However, the number of participants should be more justified.

-          Line 113: what do you mean by “pre-post-intervention design” ?

-          Please include more specifications of the DeXtreme robot and the associated serious game.

-          Results are good and well discussed.

-          The Conclusion should be added by integrating the limitations and the perspective.

Reviewer 3 Report

Comments and Suggestions for Authors

This study is of interest given return of UE and hand function is so limited post  recovery from a significant stroke with  measurable hemiplegia.  The use of robotic training with feedback allows more repetitions than may be tolerated without motivating and engaging feedback.   The findings and results are positive  for some of the dependent variables following only 5 hours of training. 

     The authors address some of the weaknesses of the study in the discussion section. What they address is relevant.  However, I. believe a few additions are needed before this manuscript should. be published. 

    1.  More limitations need to be stated:  a. no control group to account for  equal time with patient engagement even without the robot b.  were small statistical gains actually correlated with improved function  c.  How could they have improved the demand for hand function versus arm movement. 

 2.  Authors need to calculate effect size for the changes.  The amount of change was small even when statistically significant.  It is not clear to me these changes were clinically relevant and clinically significant.  Calculating the effect size would be helpful 

3.  The authors should summarize what changes were not significant.  They used a lot of outcome measures and the majority of the change scores were not significant.  Even the ones that were significant were very small.  

4.  Surprising that patients did not report pain.  This is a little unusual.  The authors should reference some other studies where post stroke patients did not have pain.
